

# PLD2 regulates microtubule stability and spindle migration in mouse oocytes during meiotic division

Xiaoyu Liu, Xiaoyun Liu, Dandan Chen, Xiuying Jiang and Wei Ma

Department of Histology and Embryology, School of Basic Medical Sciences, Capital Medical University, Beijing, China

Corresponding author
Wei Ma, mawei1026@ccmu.edu.cn

## ABSTRACT

Phospholipase D2 (PLD2) is involved in cytoskeletal reorganization, cell migration, cell cycle progression, transcriptional control and vesicle trafficking. There is no evidence about PLD2 function in oocytes during meiosis. Herein, we analyzed PLD2 expression and its relationship with spindle formation and positioning in mouse oocyte meiosis. High protein level of PLD2 was revealed in oocytes by Western blot, which remained consistently stable from prophase I with intact germinal vesicle (GV) up to metaphase II (MII) stage. Immunofluorescence showed that PLD2 appeared and gathered around the condensed chromosomes after germinal vesicle breakdown (GVBD), and co-localized with spindle from pro-metaphase I (pro-MI) to metaphase I (MI) and at MII stage. During anaphase I (Ana I) to telophase I (Tel I) transition, PLD2 was concentrated in the spindle polar area but absent from the midbody. In oocytes incubated with NFOT, an allosteric and catalytic inhibitor to PLD2, the spindle was enlarged and center-positioned, microtubules were resistant to cold-induced depolymerization and, additionally, the meiotic progression was arrested at MI stage. However, spindle migration could not be totally prevented by PLD2 catalytic specific inhibitors, FIPI and 1-butanol, implying at least partially, that PLD2 effect on spindle migration needs non-catalytic domain participation. NFOT-induced defects also resulted in actin-related molecules' distribution alteration, such as RhoA, phosphatidylinosital 4, 5-biphosphate (PIP2), phosphorylated Colifin and, consequently, unordered F-actin dynamics. Taken together, these data indicate PLD2 is required for the regulation of microtubule dynamics and spindle migration toward the cortex in mammalian oocytes during meiotic progression.

## INTRODUCTION

Meiotic maturation of oocyte is characterized by two rounds of accurate segregations of genome and asymmetric divisions of cytoplasm, resulting in two tiny polar bodies and one large oocyte. The asymmetric division is critical for the oocyte to store as much material stores as possible for the latter embryo development (*Maro & Verlhac, 2002*). The asymmetric division firstly depends on the proper location of the spindle (*Gonczy, 2002*). In mitotic animal cells, astral microtubules interact with the cortex to position the

cleavage furrow, however, in meiotic oocytes, which are absent of centrioles and astral microtubules (*Almonacid & Paoletti, 2010*; *Von Dassow, 2009*), a substitution mechanism is established to locate the spindle position. In mouse oocytes, spindle migration is an actin-dependent event rather than a microtubule-associated mechanism. Microtubule depolymerization with nocodazole treatment did not affect the faster cortex-towards movement of chromosomes (*Li et al., 2008*); however, either the stabilization of actin polymerization by jasplakinolide or its depolymerization by cytochalasin B impeded the migration of the spindle and the chromosomes group to the cortex (*Sathananthan et al., 2006*; *Li et al., 2008*).

The actin meshwork in consistent remodeling state is needed in order to 'move' the spindle, which has been ascertained by the visualization of actin with developed probes in oocytes alive. The actin network density is initially high at prophase I, then dropped shortly around germinal vesicle breakdown (GVBD), but gradually increased during spindle migration to the cortex during meiosis I progression (*Chaigne, Verlhac & Terret, 2012*). The actin dynamic is regulated by actin nucleators, including Formin 2 (FMN2) and Arp2/3 complex. FMN2 is a 'straight' actin nucleator, accompanied with spire1/2, mainly involved in actin nucleation in cytoplasm area, and the 'branched' actin nucleater Arp2/3 dominantly regulates actin nucleation in the cortex. In addition, there are also other factors regulating actin polymerization or depolymerization, including small GTPase Rho A, phosphatidylinosital 4, 5- biphosphate (PIP2) and Colifin. It is known that Rho A regulates the reorganization of filament actin (F-actin). The reduction of PIP2 synthesis suppresses actin polymerization and motility, while increasing PIP2 synthesis promotes these activities. Colifin, known as ADF/Colifin, which splices F-actin into small pieces, will lose its splicing activity after being phosphorylated at Ser3 by several proteins, such as LIM kinase 1 (LIMK1), so the cellular level of p-Colifin$^{Ser3}$ could indirectly show the actin assemble state (*Pfender et al., 2011*; *Sun et al., 2011*).

Phospholipase D2 (PLD2), one phospholipase D isoform, contains two conserved HKD domains in C-terminus, one phox homology (PX) and one pleckstrin homology (PH) domain in N-terminus. HKD is the catalytic domain and hydrolyzes phosphatidylcholine into signal molecule phosphatidic acid (PA) and soluble choline. Emphatically with PA as a key lipid second messenger, PLD2 participates in many biological processes, such as cell proliferation, membrane trafficking, cytoskeleton reorganization and cell migration (*Bruntz, Lindsley & Brown, 2014*). Both PX and PH are regulatory domains with many novel functions recently revealed. The PX domain acts as guanine nucleotide exchange factor (GEF) transforming the small GTPase, such as Rac2 or RhoA, from GDP-bound to active GTP-bound form, and thus stands for membrane ruffling and stress fiber formation, importantly independent of the lipid activity (*Ganesan et al., 2015*; *Jeon et al., 2011*; *Mahankali et al., 2012*; *Ponting, 1996*; *Wishart, Taylor & Dixon, 2001*). PLD also interacts directly with cytoskeleton proteins, such as actin, tubulin and Colifin (*Chae et al., 2005*; *Han et al., 2007*; *Kusner et al., 2003*; *Lee et al., 2001*). PLD1/2 depletion result in actin cytoskeleton defects in macrophage cells (*Ali et al., 2013*). Not only the cytoskeleton reorganization function is reported in the physiologic cell processes; many researches also mention PLD2 involvement in the metastasis of cancer. A high level of PLD2 is found in

colorectal and breast cancers, implying its possible role in cancer invasion and metastasis (*Oshimoto et al., 2003*).

As PLD2 plays a key role in cytoskeleton reorganization in many cells and oocyte maturation is a process propelled by a series cytoskeleton dynamic changes, the role of PLD2 in the meiotic division is of great interest for us to discover. In this study, we detected stable expression of PLD2 in mouse oocytes during meiotic maturation, and its requirement for the normal assembly and peripheral migration of meiotic spindle, independent on its catalytic activity in PA production.

## MATERIALS AND METHODS

### Animal experiments ethics and feeding condition

All the animal experiments were strictly conducted following the policies and instructions of the Care and Use of Animals in Research and Teaching and approved by the Animal Care and Use Committee of Capital Medical University with the approval No AEEI-2015-119. Mice C57BL/6- male and BALB/C-female, together with their F1 off-springs were raised as the 12 h light /dark cycle in a proper humidity and temperature with enough food and water.

### Oocyte collection and culture

Ovaries were harvested from 21-day-old female F1 mice which were euthanatized with $CO_2$ at 44–48 h after intraperitoneal injection of 10 IU pregnant mare serum gonadotropin (PMSG, Beijing XinHuiZeAo Science and Technology). Cumulus cell-oocyte complexes (COCs) were released from ovary follicles by puncturing the ovaries with 25-gauge needles, and further cultured in Minimal Essential Medium (MEM) with 10% fetal bovine serum (FBS; Gibco, Gaithersburg, MD, USA) and 3 mg/ml bovine serum albumin (BSA, Sigma) in an atmosphere with 5% $CO_2$ and proper humidity at 37 °C. At 0, 2, 4, 8 and 17 h of culture, corresponding to meiotic stages at germinal vesicle (GV), germinal vesicle breakdown (GVBD), pro-metaphase I (pro-MI), metaphase I (MI) and metaphase II (MII), respectively, the cumulus cells were deposed and oocytes were processed for different experiments. As for the drug treatment, denuded oocytes were used and departed into control and drug- MEM and IVM to different periods and ready for other procedures.

### Pharmacological inhibition of PLD2

In order to analyze the potential function of PLD2 in mouse oocytes, three PLD2 inhibitors were used in this study, including PLD1/2 dual inhibitor 5-Fluoro-2-indolyl des-chlorohalopemide (FIPI) (sc-300694; Santa Cruz Biotechnology, Dallas, TX, USA), and the specific PLD2 inhibitor, N-[2-[1-(3-Fluorophenyl)-4-oxo-1, 3,-8-triazaspiro [4.5] dec-8-yl] ethyl]-2-naphthalenecarboxamide (NFOT) (4171, TOCRIS Bioscience), and 1-butanol (537993; Sigma-Aldrich, St. Louis, MO, USA). FIPI binds at S757 of PLD2, which is within the HKD2 catalytic site of the enzyme, rapidly blocking *in vivo* PA production with subnanomolar potency. NFOT is mixed-kinetics inhibitor, binding to PLD2 at two different sites, one being at S757/S648 in HKD domain, and another being an allosteric site that is a natural phosphoinositide biding pocket in PH domain and usually occupied

by PIP2. NFOT affects both PA production and PIP2 binding to PLD2. And 1-butanol is another PLD inhibitor, blocking PLD production of PA, and widely employed to identify PLD/PA-driven processes.

All solid drugs were reconstructed to stock solution at 50 mM in dimethyl sulfoxide (DMSO; Sigma, St. Louis, MO, USA), and further diluted in oocyte culture medium to final working concentration before use. For control, same amount of DMSO was added in the medium. The quantity of DMSO was not more than 0.1% (v/v) in working solution. 1-butanol was resupplied every hour for the reduced effect.

### Cooling treatment

Oocytes at GV stage were pre-cultured for 8 h in maturation medium supplemented with DMSO or NFOT, at which time, the majority oocytes were supposed to develop to MI stage. After thoroughly washed, these oocytes were placed on ice and incubated in fresh M2 medium (M5910; Sigma-Aldrich, St. Louis, MO, USA) for additional 20 min, 40 min and 60 min, respectively, and then processed for analysis of microtubules and PLD2 through immunofluorescence procedure.

### Immunofluorescence

Oocytes were fixed in 1% paraformaldehyde (PFA) in PEM buffer (100 mM Pipes, 1 mM $MgCl_2$ and 1 mM EGTA, pH 6.9) with 0.5% Triton X-100 for 30 min at room temperature. After triple washing in phosphate-buffered saline (PBS) containing 0.2% Triton X-100 (PBST), oocytes were blocked in PBST with 10% normal goat serum, 1% BSA and 0.3 M glycine for 1 h at room temperature, and incubated in properly diluted primary antibodies, including rabbit anti-PLD2 (1:1000, HPA013397; Sigma-Aldrich, St. Louis, MO, USA), mouse anti-acetylated tubulin (1:3000, T7451; Sigma-Aldrich, St. Louis, MO, USA), mouse anti-PIP2 (1:500, sc-53412; Santa Cruz Biotechnology, Dallas, TX, USA), mouse anti-phosphor Colifin (Ser3) (1:1000, GTX50199; GeneTex, Irvine, CA, USA) and mouse anti-RhoA (1:500, sc-418; Santa Cruz Biotechnology, Dallas, TX, USA), at 4 °C, overnight. After triple washing in PBST, each for 10 min, oocytes were treated with goat anti-mouse Alexa-488 (1:500; Molecular Probes) or goat anti-rabbit Alexa-594 (1:500; Molecular Probes) in a light-proof box for 45 min at room temperature. After washed as described, oocytes were mounted on the glass slides with Vectashield mounting medium containing DAPI (H-1200; Vector Laboratories). All images were taken by DP-97 Olympus microscope.

The actin staining was carried out following special procedure as described previously (*Na & Zernicka-Goetz, 2006*). Oocytes were exposed to the acidic Tyrode's solution (T1788, Sigma) for 2 min to remove the zona pellucida. After short recovery in culture medium, these oocytes were fixed in the fix solution (4% PFA, 0.15% glutaraldehyde, 0.06% Triton X-100, 130 mM KCl, 25 mM HEPES and 3 mM $MgCl_2$, pH 6.9), for 30 min at room temperature, and then permeabilized in 0.2% Triton X-100 in PEM buffer prior to additional 10 min incubation in fix solution. After thoroughly washed, these cells were labeled with Alexa flour 555—phalloidin (1:3000, 8953; Cell Signaling Technology, Danvers, MA, USA). The samples were mounted and analyzed as described above.

## Western blot

50 oocytes in each sample were collected in Laemmli lysis buffer (161-0737; Bio-Rad, Hercules, CA, USA) with protease inhibitor cocktail (P2714; Sigma-Aldrich, St. Louis, MO, USA) and stored at $-80\,°C$. Before use, the samples were degenerated at $100\,°C$ for 5 min and cooled down on ice. The proteins were separated by 10% SDS-PAGE and blotted to PVDF membrane (IPVH00010; Millipore, Billerica, MA, USA), the membranes were then blocked in 5% non-fat milk in Tris-buffered saline (TBS) containing 0.1% Tween-20 (TBST) for 1 h at room temperature, and then incubated overnight at 4 °C in diluted primary antibodies. After washed three times, each for 15 min, the membranes were further labeled with horseradish peroxidase-conjugated secondary antibodies (ZSGB-BIO) for 45 min at room temperature. After triple washes as described, the membranes were treated with enhanced chemiluminescence (ECL) system (P1010; Applygen Technologies Inc., Beijing, China). The semi-quantitative gray scale analysis of bands was processed using Image J software.

## Data statistics

All experiments were performed at least 3 replicates and 30–50 oocytes were included in each group. The statistics and graphs were processed and produced by GraphPad Prism 5.01 (GraphPad Software, La Jolla,CA, USA). Data were given as the mean $\pm$ SEM and $P < 0.05$ was considered significant.

## RESULTS

### Stable expression of PLD2 and its relationship with spindle in oocyte meiosis

The protein expression of PLD2 in mouse oocytes was firstly detected by Western blot analysis. As shown in Fig. 1, a high level of PLD2 protein expression was revealed in mouse oocytes; importantly, the protein level was consistently stable from germinal vesicle (GV) stage to metaphase II (MII) (Fig. 1A), and further statistical analysis of the band gray scale confirmed there was no significant difference in PLD2 level among different meiotic stages, as checked at GV, germinal vesicle breakdown (GVBD), metaphase I (MI) and MII stage, respectively (Fig. 1B).

Then, the subcellular distribution pattern of PLD2 was explored with an immunofluorescence approach. As shown in Fig. 1, no special aggregation was observed in oocytes at GV stage (Fig. 1C, 3), indicating PLD2 was evenly distributed in cytoplasm before the resumption of meiosis. Upon GVBD, as the chromatin was condensed into individual chromosomes (Fig. 1C, 5), PLD2 began to aggregate around the condensing chromosomes, simultaneously with the newly formed microtubules (Fig. 1C, 6–8). As cell cycle progressed to pro-metaphase I (pro-MI) and MI stage, microtubules were gradually organized into bi-polar spindle with all the chromosomes properly aligned on the equatorial plate, evidently, PLD2 sustained its co-localization with microtubules throughout the whole process of spindle organizing (Fig. 1C, 10–12, 14–16). The co-localization pattern lasted to the coming anaphase, during which PLD2 was overlapped with microtubules at polar area (Fig. 1C, 18–20). When oocytes developed to MII stage, PLD2 was again distributed

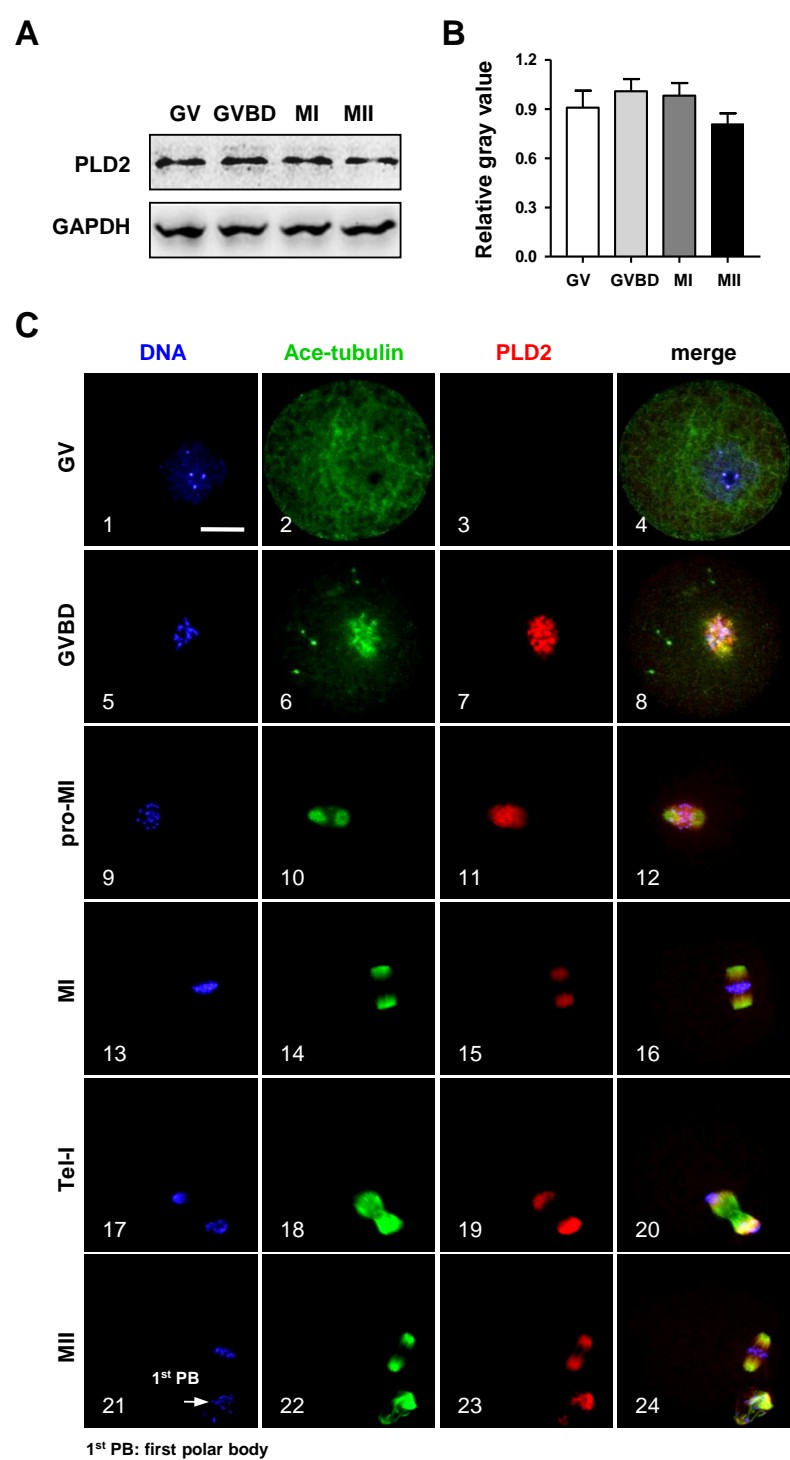

**Figure 1 The protein expression of PLD2 and its relationship with spindle in mouse oocytes during meiotic maturation.** (A) Western blot analysis detected stable expression of PLD2 at the GV, GVBD, MI and MII stages, respectively. The experiments were repeated triply. (B) Statistical analysis confirmed no significant difference in PLD2 expression among germinal vesicle (GV), germinal vesicle breakdown (GVBD), metaphase I (MI) and metaphase II (MII) stages ($P > 0.05$). 

**Figure 1 (...continued)**
(C) PLD2 was co-localized with spindle during meiotic division. At the GV stage, no particular aggregation of PLD2 was detected throughout the cytoplasm and nuclear (1–3). After GVBD, PLD2 emerged as filamentous assembly and was co-localized with newly formed microtubules around the condensed chromosomes (5–8). From pro-MI to MI, PLD2 was co-localized with the spindle structure (9–16), and during anaphase I (AI) to telophase I (Tel I), it was mainly distributed at spindle poles and absent from the midbody area (17–20). At the MII stage, PLD2 was again co-localized with microtubules on the re-formed meiotic spindle and first polar body (1st PB) (21–24: arrow). DNA was visualized in blue, microtubules were in green and PLD2 was in red. Scale bar = 20 μM.

across the re-formed spindle and precisely co-localized with microtubules (Fig. 1C, 21–24). These data strongly suggest possible involvement of PLD2 in meiotic spindle formation or positioning in mouse oocytes.

## PLD2 inhibition suppressed spindle migration to the cortex during oocyte meiosis

We intended to explore the possible role of PLD2 in oocyte meiotic maturation by using a specific inhibitor to PLD2. After GV oocytes were cultured for 17 h with NFOT at 20 μM, which was a defined proper concentration after series of experiments, nearly all the oocytes were arrested at MI stage but not matured to MII, further statistical analysis demonstrated that the number of oocytes with first polar body extrusion was significantly lower in the NFOT group than that in the control (Fig. 2B). Importantly, all the MI oocytes were installed with an enlarged spindle which was center-positioned (Fig. 2A).

We went back to check spindle structure and positioning at 8 h, 10 h and 12 h of NFOT incubation. As showed in Fig. 2C, all the oocytes remained at MI stage in NFOT group (Fig. 2C, 1′–3′), no matter the incubation time extended up to 12 h, and the spindle was enlarged and located closely to the central area (Fig. 2C, 4′–6′). PLD2 was co-localized with microtubule on spindle and aggregated as big dots, distributed near to spindle poles (Fig. 2C, 7′: arrows) and randomly in cytoplastic area (Fig. 2C, 7′: arrowhead). In contrast, nearly in all control oocytes, the spindle exhibited normal size and was located in the sub-cortex position (Fig. 2C, 4–5), some oocytes developed to MII stage as culture time increased to 12 h (Fig. 2C, 3). To quantitatively compare spindle changes after NFOT treatment, we measured the value of long axis length of spindle (l), oocyte spherical radius (R) and the distance between spindle center and oocyte spherical center (d) in each oocyte. The ratio of "l" to "R" indicated the relative value of spindle length and was denoted as "L" (L, L = l / R). The ratio of "d" to "R" indicated the relative value of spindle migrating distance and was denoted as "D" (D, D = d/R) (Fig. 2D). Statistical analysis demonstrated that the "L" value was significantly higher in NFOT-treated oocytes than that in control at 8 h, 10 h and 12 h incubation (Fig. 2E) ($P < 0.05$), indicating the spindle was dramatically enlarged with PLD2 inhibition, at the same time, the "D" value was markedly decreased in NFOT group at all the time points (Fig. 2F) ($P < 0.001$), implying spindle migrating toward the cortex was suppressed in oocytes when PLD2 was inhibited with NFOT.

As the enlarged spindle, we proposed that suppression of PLD2 may change the organizing pattern of spindle microtubules, and this proposal was proved by the classic cold-induced microtubule depolymerization experiment. When MI oocytes were incubated

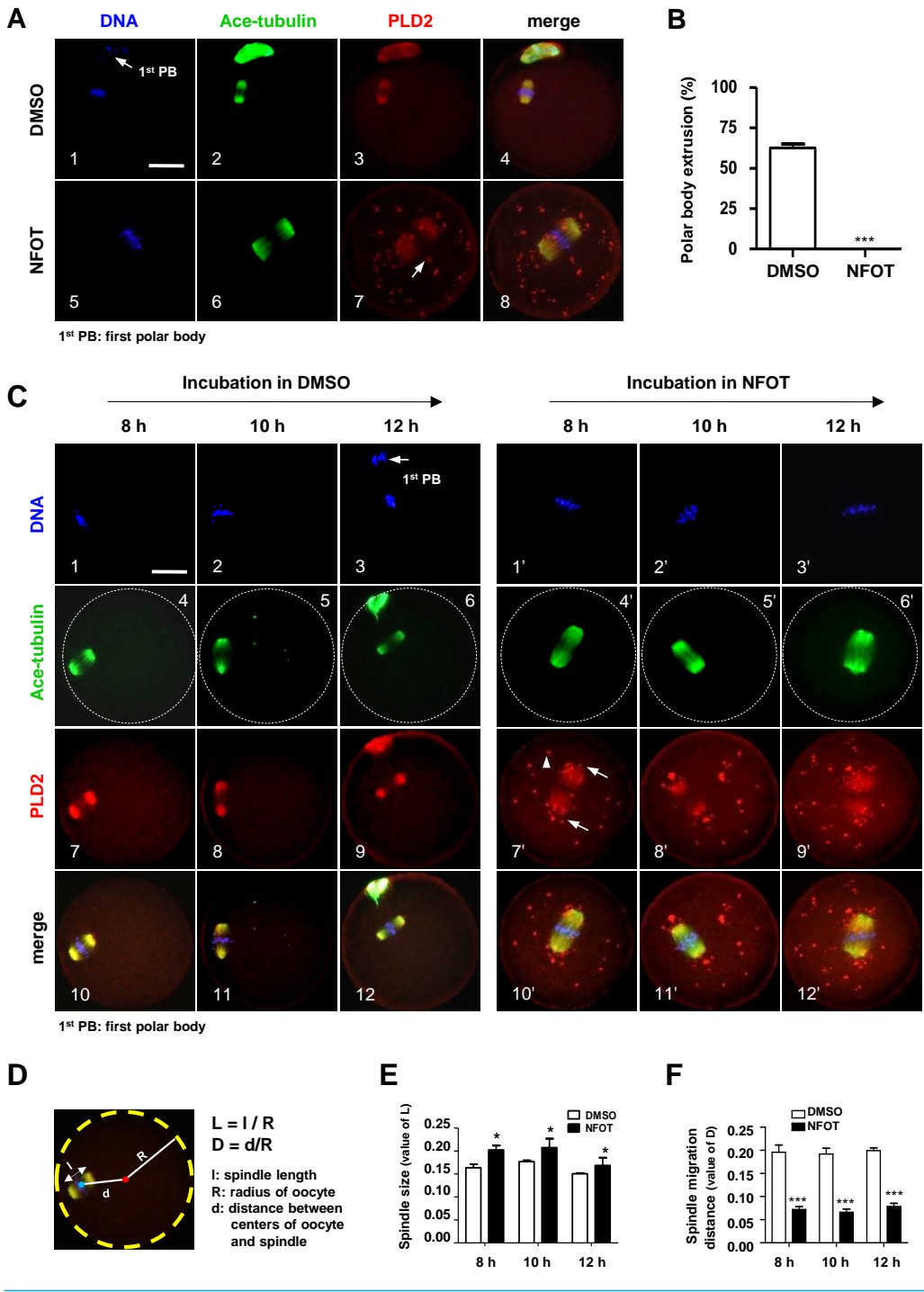

**Figure 2** **PLD2 inhibition resulted in center-positioned spindle and arrested meiotic progression.** GV oocytes were *in vitro* cultured for 8 h, 10 h, 12 h and 17 h, respectively, in maturation medium with 20 mM NFOT, and then collected for immunofluorescence analysis. Oocytes incubated with only DMSO was used as control. The experiment was repeated at least three times with more than 30 oocytes analyzed in each group every time. DNA was labeled in blue, microtubules were in green and PLD2 was in red. Scale bar = 20 μM. (continued on next page...)

**Figure 2 (…continued)**
(A) Representative images showed center-positioned spindle with enlarged size and meiotic arrest at MI after 17 h incubation in NFOT (5–8), PLD2 was labeled on spindle and also exhibited as big dots located in the polar area and cytoplasm (7: arrow). Control oocytes developed to MII stage with normal spindle size and PLD2 distribudtion (1–4, arrow: first polar body, 1st PB). (B) Result of statistical analysis demonstrated the number of MII oocytes, manifested with extruded 1st PB, was significantly lower in NFOT group after 17 h incubation ($P < 0.001$). (C) Representative images showed spindle status and meiotic progression in oocytes when checked at 8 h, 10 h and 12 h (4'–6'), in NFOT-treated oocytes, meiotic progressed was arrested at the MI stage with enlarged spindle nearly at the center area, PLD2 was labeled on spindle and also as large dots in area near to spindle poles (7': arrows) and across the cytoplasm (7': arrowhead). In oocytes incubated with DMSO alone, the spindle was assembled in normal size and positioned in the cortex (4, 5), PLD2 was co-localized with microtubules on spindle (7–9), some oocytes were progressed to MII stage when checked at 12 h (3, 6, 8, 12: arrow: 1st PB). (D) Computational method for measurement of spindle size and migration distance. A red dot indicates the spherical center of oocytes and a blue dot indicates the center of spindle, "l" indicates the length of spindle long axis, "R" indicates the spherical radius of oocytes, and "d" indicates the distance from spindle center to spindle center; "L" is the ratio value between "l" and "R" (L = l/R), representing the relative length of spindle structure, "D" is the ratio value between "d" and "R", indicating the relative distance of spindle migration. "l", "R", and "d" values were obtained using Image-Pro Plus 6.0 software. (E) The results of statistical analysis indicated the spindle size ("L" value) was significantly bigger in NFOT-treated oocytes than that in control when analyzed at 8 h, 10 h and 12 h of drug incubation ($P < 0.05$). (F) Statistical data indicated the distance of spindle migrating to the cortex ("D" value) was significantly smaller in NFOT-treated oocytes than that in control when analyzed at 8 h, 10 h and 12 h of drug incubation ($P < 0.001$).

on ice for different periods with NFOT, we found that microtubules were more stable in NFOT-treated oocytes. As showed in Fig. 3, microtubules were totally disassembled in control oocytes after 20 min incubation on ice (Fig. 3, 1), simultaneously, PLD2 assembly was also disappeared (Fig. 3, 5). However, in NFOT group, microtubules remained with PLD2 in polar area at 20 min of incubation (Fig. 3, 2, 6), even detectable at 40 min (Fig. 3, 3, 7), and totally depolymerized when checked at 60 min (Fig. 3, 4, 8). Microtubule dynamics was highly consistent with PLD2 signal during the whole process of cold treatment. These data indicate that PLD2 is associated with normal stability of microtubules, and PLD2 inhibition could promote microtubule stability, might contributing to the formation of large spindle.

## Catalytic inhibition of PLD2 did not affect the cortex-towards spindle movement

PLD2 catalyzes the hydrolysis of phosphatidylcholine to produce PA and choline, and this function is dependent on PLD2 catalytic domain HKD. NFOT is an allosteric and catalytic inhibitor, targets two different sites in HKD and PH domain, suppressing both PA production and PIP2 binding to PLD2. FIPI is specific catalytic inhibitor to PLD and binds to HKD domain, inhibiting PA production (*Monovich et al., 2007*). The alcohol 1-butanol also blocks PLD-catalyzed PA generation (*McDermott, Wakelam & Morris, 2004*). In order to clarify whether PLD2 catalytic activity was involved in spindle migration, oocytes were treated with two catalytic inhibitors, FIPI and 1-butanol, in the same manner of NFOT. Beyond our expectation, we were astonished to find that FIPI nearly exerted no effect on spindle migration even in a cell-death dose (Fig. 4). Similarly, 0.1% 1-butanol did not disrupt the spindle movement but indeed enlarged the size of spindle (Fig. 5). There data

**Incubation on ice**

| DMSO | NFOT | | |
|---|---|---|---|
| 20 min | 20 min | 40 min | 60 min |

**Figure 3** **The spindle microtubules were stabilized in oocytes with PLD2 inhibition with NFOT.** GV oocytes were incubated for 8 h with DMSO or NFOT, and then collected for additional incubation in fresh M2 medium on ice for 20 min, 40 min and 60 min, respectively, and followed by immunofluorescence procedure. The experiment was repeated triply with more than 30 oocytes analyzed in each group every time. Microtubules were labeled in green and PLD2 in red. Scale bar = 20 μM. At 20 min of cooling treatment, the microtubules were totally disassembled together with PLD2 in control oocytes; however, both microtubules and PLD2 remained in NFOT-treated oocytes at this time, even sustained at 40 min, and completely vanished at 60 min.

inferred that PLD2 HKD domains probably do not take part in the regulation of spindle cortex-towards migration.

## PLD2 suppression altered actin dynamics in mouse oocytes

As the spindle is disabled in movement under the conditions of the PLD2 inhibition with NFOT, we went back to assay to see if any changes occurred in the actin network and the relative several proteins, such as PIP2 and Colifin. With the specific method of actin fixation, we found that actin was presented in a fibro-crossing manner throughout the cytoplasm in control oocytes; however, in the NFOT treatment group, actin distribution changed to forming many clusters quite big in size on the oocyte surface, with no obvious actin fiber crossover in cytoplasm (Fig. 6). It is plausible that PLD2 inhibition by NFOT resulted in great change in oocyte actin distribution, which led to the disable movement of spindle from accentor to the cortex.

Since PIP 2 and Colifin are tightly associated with actin polymerization and dynamics, we continued to examine the cellular level of these proteins in NFOT-treated oocytes. By immunofluorescence in normal MI oocytes, we found that both PIP2 and phosphorylated Colifin (p-Colifin[Ser3]) were aggregated as bright foci at the poles of spindles (Fig. 7A, 2;

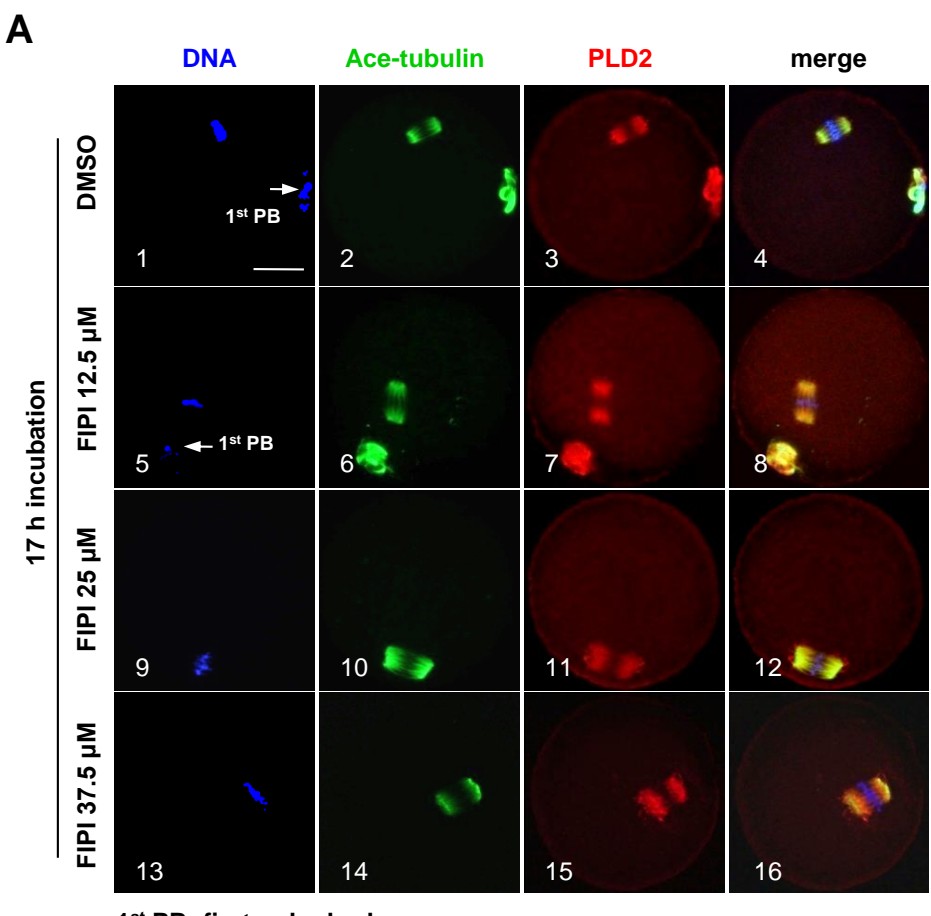

**1st PB: first polar body**

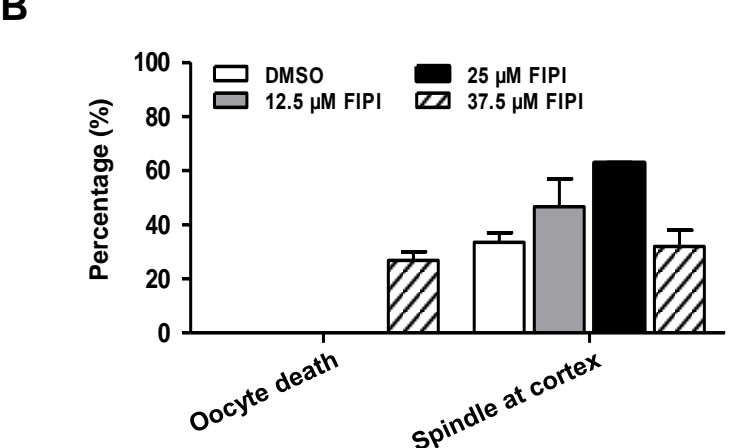

**Figure 4** **PLD catalytic inhibitor FIPI exerted no effect on spindle migration.** GV oocytes were cultured for 17 h in maturation medium with DMSO, 12.5 μM FIPI, 25 μM FIPI and 37.5 μM FIPI, respectively, and then fixed for immunofluorescence analysis. The experiment was repeated at least three times. DNA was labeled in blue, microtubules in green and PLD2 in red. Scale bar = 20 μM. (A) Representative images illustrated meotic progression and spindle state after drug incubation. (continued on next page…)

**Figure 4 (…continued)**
The majority oocytes progressed to MII stage in DMSO, 12.5 µM FIPI and 25 µM FIPI group, manifested with the extruded 1st PB (1, 5: arrow), the MI oocytes in these groups contained typical bi-polar spindles, which were organized in normal size and positioned in the cortex area (10). Nearly 27% oocytes were dead in 37.5 µM FIPI group, but the rest MI oocytes were equipped with spindles in normal size and position (14–16). (B) Statistical data indicated there was no significant difference in the number of MI oocytes containing spindle in normal size and cortical position among all the treatment groups.

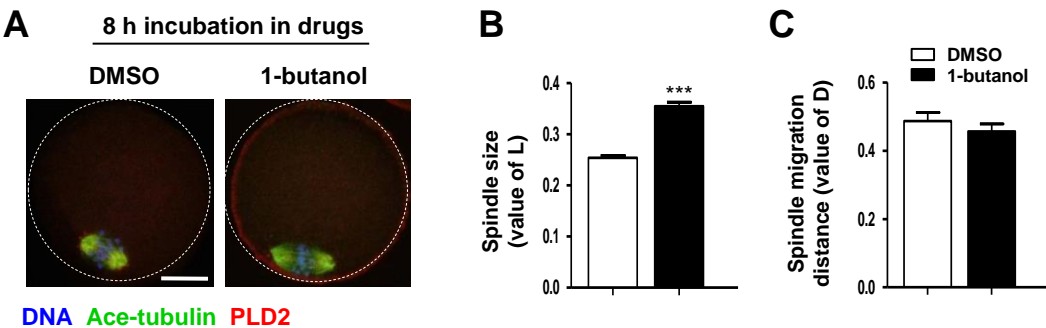

**Figure 5  1-butanol did not affect spindle cortex migration.** Oocytes at GV stage were cultured for 8 h in maturation medium containing 0.1% 1-butanol, and then fixed for immunofluorescence analysis. Three replicates were included in this experiment, and at least 30 oocytes were analyzed in each group every time. DNA was stained in blue, microtubules in green and PLD2 in red. Scale bar = 20 µM. (A) Typical images illustrated the spindle was assembled in large size but positioned in the cortex after 1-butanol incubation. (B) Statistical data showed that the spindle was significantly enlarged in 1-butanol group than in control ($P < 0.001$). (C) Results of statistical analysis confirmed 1-brutanol had no effect on spindle migration toward the cortex.

Fig. 7B, 3: arrows) and also randomly in cytoplasm area (Fig. 7A, 2; Fig. 7B, 3: arrowheads), just like the signal of microtubule orgalizing center (MTOC); however, the foci signal of PIP2 definitely disappeared in oocytes incubated with NFOT (Fig. 7A, 6); similarly, p-Colifin[Ser3] signal was also significantly weakened or totally disappeared in NFOT-treated oocytes (Fig. 7B, 7). Further statistical analysis confirmed that the number of oocytes with bright foci of PIP2 or p-Colifin[Ser3] was significantly reduced after NFOT treatment (Figs. 7C and 7D) ($P < 0.001$). Consistent with this Western blot analysis indicated that the p-Colifin[Ser3] protein level was reduced greatly while the PLD2 level remained stable (Fig. 7E), and this was supported by statistical data that the band gray intensity of p-Cofilin[Ser3] was significantly decreased in NFOT-treated oocytes (Fig. 7F). The two weakened signals indirectly indicate that at least at the spindle poles the actin was more likely to be in a disassembly state under the condition of PLD2 inhibition, which may stand for the unmoved spindles.

In addition, we found the small GTPase RhoA, which can bind with PLD2 and regulate actin dynamics, was organized into a bi-polar structure with PLD2 localized mainly at the polar area in MI oocytes (Fig. 8). In NFOT-treated oocytes, RhoA sustained in "spindle-like" assembly, and also concentrated as additional dots in the cytoplasm, which were co-localized with the cytoplasmic PLD2 dots (Fig. 8A, 6–8: arrows). Results of statistical analysis confirmed the proportion of oocytes with cytoplasmic RhoA dots was pronouncedly higher than that in control (Fig. 8B) ($P < 0.001$). The changed distribution

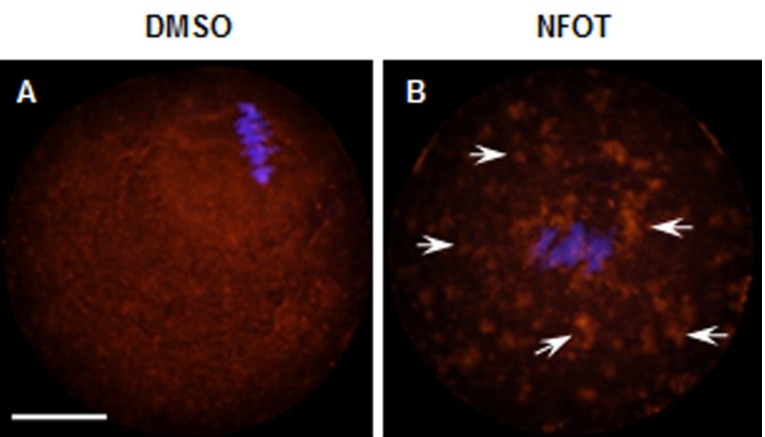

**DMSO** **NFOT**

Actin (phalloidin)  DNA (DAPI)

**Figure 6 Cytoplasmic distribution of actin was changed in NFOT-treated oocytes.** Oocytes were labeled with Alexa flour 555—phalloidin and DAPI after 8 h incubation with DMSO or NFOT. The experiment was repeated three times. DNA was visualized in blue and actin in red. Scale bar = 20 μM. (A) In DMSO group, the actin was assembled as quite thick fibers in the cytoplasm, with pretty high concentration surrounding the spindle area (arrows on the left), while in the NFOT group, fibers were taken place by the big clusters (arrows on the right). (B) Statistical analysis confirmed that the proportion of oocytes with actin clusters in cytoplasm was significantly higher in NFOT group ($P < 0.001$).

of RhoA must affect its function in regulation of actin polymerization and dynamics. NFOT-induced big dots in the cytoplasm are newly aggregated structure, may have some relationship with some transport-vesicles in the cytoplasm such as Golgi stacks and endosomes.

## DISCUSSION

The present study shows that PLD2 is consistently expressed in mouse oocytes during meiotic maturation and co-localized with microtubules on spindle structure. PLD2 inhibition can affect microtubule stability and suppress spindle migration to the cortex, through regulating some actin-related factors, but independent of its catalytic activity.

We suppressed PLD2 activity by a specific inhibitor NFOT and found that the meiotic progression was impeded as the enlarged spindle remained in the center and was unable to approach the cortex. By using FIPI, which inhibited both PLD1/2 by inhibit the catalytic (HKDs) domains, we were surprised to discover that spindle movement was not much impaired even in a lethal dose. Again, 1-butanol, which cuts off the production of PA, did not affect the peripheral migration of spindle. As NFOT has dual-inhibited sites on PLD2, one in the regulatory domain, and the other in the catalytic domain, we supposed that PLD2 may take part in the spindle migration through a PA-independent way, in other words, not by its catalytic domain. Although many researchers have shown that the PA is a key lipid second messenger in many biological processes, we probably proved the rest roles of PLD2.

In oocytes with PLD2 inhibition, the spindle was obviously enlarged, suggesting normal PLD2 activity may be associated with mechanism responsible for spindle size maintenance.

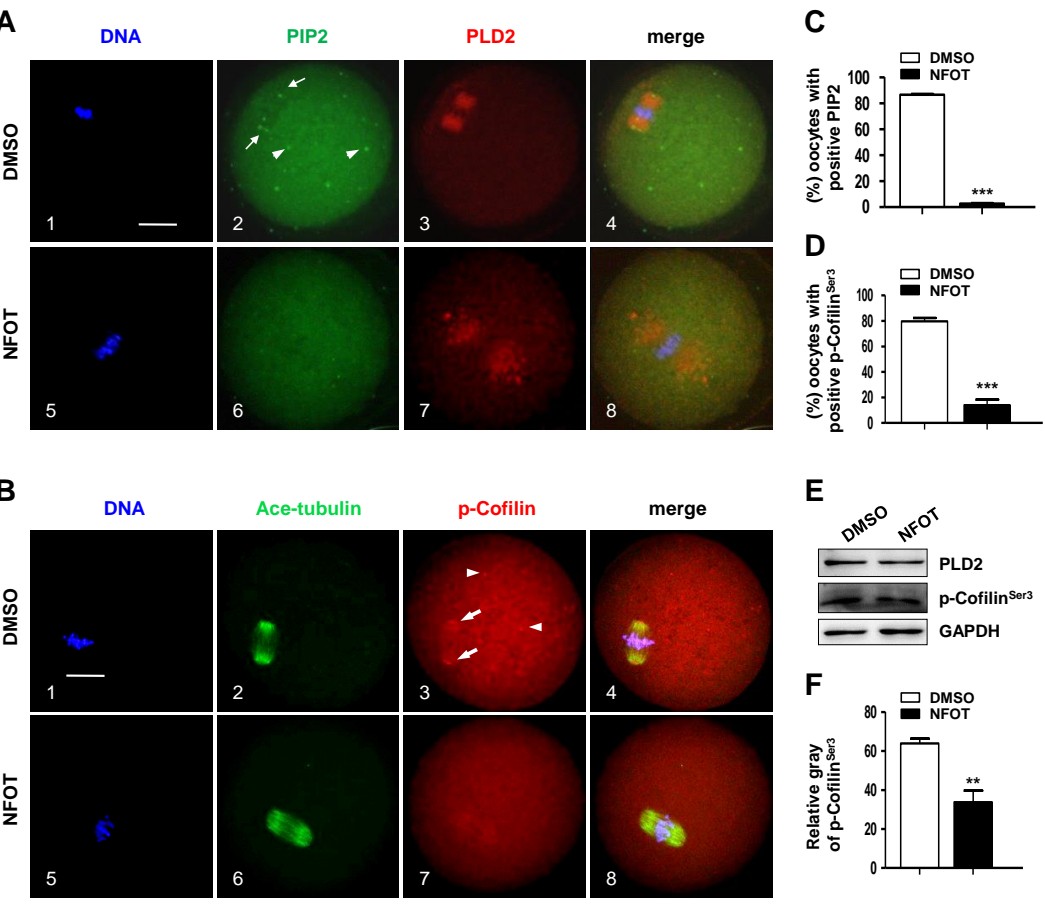

**Figure 7  PLD2 inhibition destroyed polar recruitment of PIP2 and p-Cofilin$^{Ser3}$.** GV oocytes were cultured for 8 h in maturation medium with 20 mM NFOT, and then fixed for analysis of PIP2 and p-Cofilin$^{Ser3}$. The experiment was repleated at least three times, and 100–150 oocytes were included in each group every time. DNA was stained in blue, microtubules and PIP2 in green, and PLD2 and p-Cofilin$^{Ser3}$ in red. Scale bar = 20 $\mu$M. (A) Immunofluorescence showed PIP2 was concentrated on spindle polar area (2: arrows) and distributed as bright foci in the cytoplasmic area (2: arrowheads) in control oocytes, in contrast, PIP2 localization on spindle poles and distribution in cytoplasm was dramatically reduced or completely disappeared in NFOT-treated oocytes (6). (B) Typical images illustrated that p-Cofilin$^{Ser3}$ was assembled as bright foci, which were localized on the poles of spindle (3: arrows) and also randomly distributed in the cytoplasmic area (3: arrowheads) in control oocytes; however, p-Cofilin$^{Ser3}$ aggregation on spindle poles and in cytoplasm was depleted in NFOT-treated oocytes (7). (C) Statistical data showed that the number of oocytes with bright PIP2 signal was significantly reduced in NFOT-treated oocytes than in control ($P < 0.001$). (D) Statistical analysis confirmed that the proportion of oocytes with bright p-Cofilin$^{Ser3}$ was markedly decreased in NFOT group than in control ($P < 0.001$). (E) Western blot analysis showed the protein expression of p-Cofilin$^{Ser3}$ was reduced in NFOT-treated oocytes, meanwhile, PLD2 expression was not affected. GAPDH expression was also detected and employed as loading control. (D) Statistical analysis of blot band gray indicated p-Cofilin$^{Ser3}$ protein level was significantly lower in NFOT group than in control ($P < 0.01$).

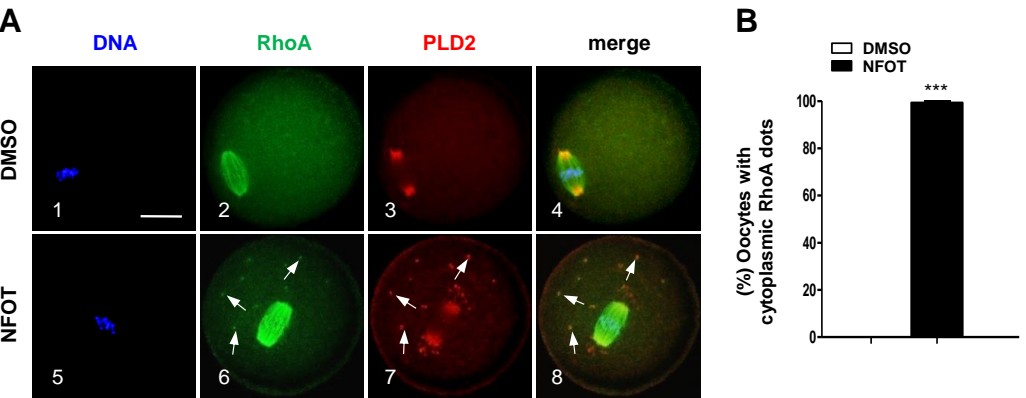

**Figure 8  NFOT treatment changed RhoA cytoplasm distribution.** Oocytes at GV stage were incubated for 8 h with or without NFOT, and then fixed for immunostaining of PLD2 and RhoA. This experiment was repeated three times, with 50 oocytes included in each group every time. DNA was labeled in blue, RhoA in green and PLD2 in red. Scale bar = 20 µM. (A) Immunofluorescence showed co-localization of RhoA with PLD2 was only detected in the spindle area in control cells (2–3); however, in NFOT-treated oocytes, RhoA was co-localized with PLD2 in the spindle area, but also aggregated as foci and overlapped with PLD2 dots in cytoplasm (6–8: arrows). (B) Statistical analysis confirmed the proportion of oocytes with cytoplasmic RhoA dots was significantly higher in NFOT group ($P < 0.001$).

In mammalian oocytes, mitogen-activated protein kinase (MAPK) activity regulates microtubule organizing and spindle length (*Sun, Breitbart & Schatten, 1999*). Its family member p38α MAPK cooperates with kinesin molecule Eg5 to maintain the scale magnitude of spindle structure in oocytes, the former works to hold the spindle magnitude while the latter functions just the opposite. The depletion of p38α MAPK could induce spindle elongation and meiotic arrest in mouse oocytes (*Ou et al., 2010*), this phenotype is similar with those in PLD2-inhibited oocytes. PLD2 has been proved to be involved in activating the MAPK pathway (*Rizzo et al., 1999*). The signal crossroad PLD2—PA—diacylglycerol (DAG) mediates the activation of p38-MAPK (*Yang et al., 2010*), and PLD2 also activates Ras-MAPK-lL-2 in T cells (*Hamdi et al., 2008*). This evidence implies that the PLD2 and MAPK pathways may work together to regulate spindle assembly in oocytes; whether spindle length maintenance in the PLD2 downstream effect is mediated by MAPK requires further experimentation.

It is well known that spindle relocation relies on actin networks in mouse oocytes, and we searched to find proteins or mediators that not only regulate actin dynamics but also have potential interaction with PLD2, especially the none-catalytic domain, conveying the effect onto the actin. It has been indicated for years that PLD2 is a GEF for activating small GTPase RhoA (*Gomez-Cambronero, 2011*; *Jeon et al., 2011*), and a previous study shows that RhoA can regulate actin dynamics during porcine oocyte maturation (*Zhang et al., 2014*). In the present study, we found that PLD2 could interact directly with RhoA in mouse oocytes and in the condition of PLD2 inhibition, both PLD2 and RhoA were aggregated into big dots in the cytoplasm in addition to co-localization with microtubules on the spindle, suggesting RhoA as a potential candidate that mediates the PLD2 effect on the actin network. PIP2 is well-known as an essential lipid messenger at

the plasma membrane, where its localized generation regulates endocytosis, exocytosis, actin cytoskeleton dynamics, focal adhesion assembly, ion channels and transporters (*Tan et al., 2015*). Recent evidences demonstrate that reduced PIP2 synthesis suppresses actin polymerization and motility, while increasing PIP2 synthesis enhances these activities (*Breitbart & Finkelstein, 2015*). PIP2 binds specifically to the PH domain of PLD isoforms with high affinity and independent of the substrate phosphatidylcholine, implying a key role of the PH domain in PLD function (*Hodgkin et al., 2000*; *Sciorra et al., 2002*), and coordination between PLD2 and PIP2 in regulating actin polymerization and function. NFOT inhibits PLD2 activity by blocking PIP2 binding with PLD2. As we found in this study, the polar location of PIP2 was disrupted in NFOT-treated oocytes, suggesting the local state of actin polymerization at spindle polar area must be altered, and which logically accounts for the delayed migration of spindle. Colifin is known to sever actin filaments by creating more positive ends on filament fragments, this actin-depolymerizing protein can be phosphorylated at Ser3 and inactivated by LIMK1, thus an increasing p-Cofilin$^{Ser3}$ level can promote actin polymerization (*Pfender et al., 2011*; *Sun et al., 2011*). It is newly revealed that Cofilin is a PIP2-binding protein in somatic cells (*Sengelaub et al., 2016*), consistent with our finding that both PIP2 and p-Colifin$^{Ser3}$ were localized at the polar area of spindle in mouse oocytes, and interestedly, this polar recruitment was destroyed by PLD2 inhibition. Collectively, the information may suggest PIP2, p-Cofilin$^{Ser3}$ and PLD2 are components of super complex at spindle poles, and individually, PLD2 serves as a scaffolding plate, PIP2 interacts with PLD2 and mediates p-Cofilin$^{Ser3}$ binding to the complex, thus these molecules work together to regulate actin polymerization and spindle migration. PLD2 dysfunction would affect the complex formation and inhibit spindle towards-cortex migration.

In summary, the data presented here suggest that PLD2 is involved in spindle formation and its migration to the cortex in mouse oocytes during meiotic progression, particularly this function is independent of PLD2 catalytic activity in phosphatidylcholine hydroxylation.

## ACKNOWLEDGEMENTS

The authors thank Dr. Qian Wang, Dr. Juan Du, Dr. Yuanjing Liang and Ms. Jing Weng for their technical help and critical reading of the manuscript.

### Funding

This study was supported by grants from the National Natural Science Foundation of China (31471108, 31271253, 31500942, 31600944 and 81671454). The funders had no role in study design, data collection and analysis, decision to publish, or preparation of the manuscript.

### Grant Disclosures

The following grant information was disclosed by the authors:

National Natural Science Foundation of China: 31471108, 31271253, 31500942, 31600944, 81671454.

## Competing Interests

The authors declare there are no competing interests.

## Author Contributions

- Xiaoyu Liu conceived and designed the experiments, performed the experiments, analyzed the data, wrote the paper, prepared figures and/or tables.
- Xiaoyun Liu performed the experiments.
- Dandan Chen contributed reagents/materials/analysis tools, reviewed drafts of the paper.
- Xiuying Jiang performed the experiments, contributed reagents/materials/analysis tools.
- Wei Ma conceived and designed the experiments, analyzed the data, wrote the paper, prepared figures and/or tables.

## Animal Ethics

The following information was supplied relating to ethical approvals (i.e., approving body and any reference numbers):

Review Board: the Animal Care and Use Committee of Capital Medical University (Approval number: AEEI-2015-119).

## Data Availability

Xiaoyu, Liu (2017): Raw data.pdf. figshare. https://doi.org/10.6084/m9.figshare.4588384.v4

Xiaoyu, Liu (2017): Raw data of spindle length and migration in oocytes treated with different concentrations of FIPI.PDF. figshare. https://doi.org/10.6084/m9.figshare.4829095.v3.

## Supplemental Information

Supplemental information for this article can be found online at http://dx.doi.org/10.7717/peerj.3295#supplemental-information.

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
