# Peer review of "PLD2 regulates microtubule stability and spindle migration in mouse oocytes during meiotic division"

_PeerJ, doi:10.7717/peerj.3295_

## Round 0.1 · original submission · Minor Revisions

Both reviewers recognize your work's merit, but they also raise some points that need your attention, especially you need to draw your conclusion and discuss your findings more properly.

Reviewer 1 ·

Basic reporting

no comment

Experimental design

no comment

Validity of the findings

no comment

Additional comments

In the manuscript entitled “PLD2 regulates microtubule stability and spindle migration in mouse oocytes during meiotic division”, Liu et al. investigated expression of PLD2 and its relationship with spindle formation and positioning during mouse oocyte meiosis. After several experiments, they figured out that PLD2 is required for regulation of microtubule dynamics and spindle migration during meiotic progression, and notably, this process is independent of PLD2 catalytic activity in phosphatidylcholine hydroxylation.
Overall, the manuscript is generally well written, but some parts still need refinements. Results are clearly demonstrated and technically sound as well, but some conclusions need to be carefully addressed based on the chemicals and concentrations they adopted. The study is interesting, however, some contexts and background were not clearly exhibited in Introduction and Discussion part, and poor discussion and insufficiency in providing significant insights into those reported mechanisms or results in oocyte meiosis dampen the impact of this study. Detailed concerns and issues based on the current manuscript are outlined below.

1. Based on the results and conclusion of this study, PLD2 is essential for proper meiosis in the mouse. However, several groups already made Knockout models, which do not show early lethal or infertile, or subfertility phenotype. Please explain and specify the possible mechanisms.

2. In this study, several chemicals were used in the experiments. It seems only one is specific, the other two are not. How to rule out those side effects or unspecific functions? In addition, how were those concentrations decided/selected when using these chemicals? Based on pilot experiments, or literature?

3. Abstract, "Immunofluorescence showed that PLD2 emerged as filaments after germinal vesicle breakdown (GVBD)", "implying PLD2 regulation of spindle is independent of its ability to catalyze the hydrolysis of phosphatidylcholine", and "may be contributed to ..." need revision.

4. "however, in meiotic oocytes, which are absent of centrosomes and astral microtubules (Almonacid & Paoletti, 2010; von Dassow et al., 2009), established a totally different mechanism to locate the spindle position" needs revision.

5. Please explain "GEF" when it first comes in "The PX domain has GEF function".

6. Section title "Experiments approbation and animal feeding regiments" needs change.

7. In this study, authors used several chemicals to treat COCs, but not denuded oocytes. What functions did cumulus cells have during this process? Whether existence of cumulus cells would change any signaling pathways? Besides, authors are also suggested to add the brief context of roles of cumulus cells in oocyte meiosis and maturation in Introduction or Discussion part, for example, sheep (PMID: 21855989), mouse (PMID: 21270427), porcine (PMID: 25442018), goat (PMID: 21453051), human (PMID: 15695316), and rat (PMID: 26679437), to explain and discuss the potential roles of cumulus cells during the treatment in the present study.

8. Please specify whether oocytes from immature mice occupy same or similar quality as in the adults? Whether 10 IU pregnant mare serum gonadotropin (PMSG) was overdosed for immature females?

9. MI to MII should be kind of fast, at most only several hours in the mouse, why authors chose "8 and 17 h of culture" with such a big gap for "metaphase I (MI) and metaphase II (MII), respectively"?

10. In this study, PLD2 localization was well studied, but the potential reasons and potential functions of these location shifting was not addressed or discussed. This shift is very similar to MAPK, which is also crucial for oocyte meiosis. In Discussion part, it seems more appropriate to discuss more on PLD2 localization and other factors that also have similar characters, such as ERK. Authors may also discuss functions of there key factors in female meiosis based on the present study, for example, in oocyte maturation (PMID: 11101281; PMID: 20948319) and oocyte activation (PMID: 21554769; PMID: 23946539), to offer more possible molecular mechanisms underlying the phenotype that had been discovered in the experiments.

11. "for additional for 20 min" needs change.

12. "PLD2 began to emerge as filamentous assembly" needs revision.

Reviewer 2 ·

Basic reporting

no comment

Experimental design

no comment

Validity of the findings

no comment

Additional comments

The authors investigated the function of PLD2 in microtubule stability and spindle migration in mouse oocytes during meiotic division. Using specific inhibitors, they proved that PLD2 is required for the regulation of microtubule dynamics and spindle migration toward the cortex in mouse oocytes. This is the first report which provided more information about the function of PLD2 in meiotic division and the authors also tried to provide more information about the underlying mechanism. This is a very interesting study and generally the manuscript is well written and very accessible.
There are still some minor issues that need to be revised to make the manuscript more reliable. Listed as below:
1. In the manuscript the authors tried to describe two phenotypes of NFOT on mouse oocyte, one is
The enlarged spindle and the other one is central location of the spindle. At the meantime, the authors tried to link these two phenotypes to different structure domains by using specific inhibitors of PLD2. More clearly explanations are needed to prove the linkage for domains, inhibitors and phenotypes.
2. The data about size of spindles in FIPI treatment experiment are needed, it seems the spindles are also enlarged as shown in Figure 4A. And if both FIPI and 1-butanol treatments showed enlarged spindles, the HDK domain may be involved in this phenotype.
3. The number of oocytes used in all experiments should be provided as well as the repeated times.
4. In line 218, The ratio of “d” to “R”…., “d” should be “I”. The manuscript should be revised carefully to avoid this kind of mistake.

---

## Round 0.2 · accepted · Accept

The authors have carefully revised the manuscript.